# Physiological and Biochemical Responses to Salt Stress of Alfalfa Populations Selected for Salinity Tolerance and Grown in Symbiosis with Salt-Tolerant Rhizobium

**Annick Bertrand [1],\*, Craig Gatzke [2], Marie Bipfubusa [3], Vicky Lévesque [1], Francois P. Chalifour [4], Annie Claessens [1], Solen Rocher [1], Gaëtan F. Tremblay [1] and Chantal J. Beauchamp [4]**

[1]  Agriculture and Agri-Food Canada, Quebec Research and Development Centre, Quebec City, QC G1V 2J3, Canada; vicky.levesque@canada.ca (V.L.); annie.claessens@canada.ca (A.C.); solen.rocher@canada.ca (S.R.); gaetan.tremblay@canada.ca (G.F.T.)

[2]  Agriculture and Agri-Food Canada, Swift Current Research and Development Centre, Swift Current, SK S9H 3X2, Canada; craig.gatzke@canada.ca

[3]  Centre de Recherche sur les Grains, Inc. (CÉROM), Saint-Mathieu-de-Beloeil, QC J3G 0E2, Canada; Marie.Bipfubusa@cerom.qc.ca

[4]  DéPartement de Phytologie, Faculté des Sciences de l'Agriculture et de l'Alimentation, 2425 rue de l'Agriculture, Université Laval, Québec, QC G1V 0A6, Canada; Francois-P.Chalifour@fsaa.ulaval.ca (F.P.C.); Chantal.Beauchamp@fsaa.ulaval.ca (C.J.B.)

\*  Correspondence: Annick.bertrand@canada.ca; Tel.: +1-418-210-5005

**Abstract:** Alfalfa and its rhizobial symbiont are sensitive to salinity. We compared the physiological responses of alfalfa populations inoculated with a salt-tolerant rhizobium strain, exposed to five NaCl concentrations (0, 20, 40, 80, or 160 mM NaCl). Two initial cultivars, Halo (H-TS0) and Bridgeview (B-TS0), and two populations obtained after three cycles of recurrent selection for salt tolerance (H-TS3 and B-TS3) were compared. Biomass, relative water content, carbohydrates, and amino acids concentrations in leaves and nodules were measured. The higher yield of TS3-populations than initial cultivars under salt stress showed the effectiveness of our selection method to improve salinity tolerance. Higher relative root water content in TS3 populations suggests that root osmotic adjustment is one of the mechanisms of salt tolerance. Higher concentrations of sucrose, pinitol, and amino acid in leaves and nodules under salt stress contributed to the osmotic adjustment in alfalfa. Cultivars differed in their response to recurrent selection: under a 160 mM NaCl-stress, aromatic amino acids and branched-chain amino acids (BCAAs) increased in nodules of B-ST3 as compared with B-TS0, while these accumulations were not observed in H-TS3. BCAAs are known to control bacteroid development and their accumulation under severe stress could have contributed to the high nodulation of B-TS3.

**Keywords:** salinity tolerance; *Medicago sativa* L.; rhizobium; nodules; amino acids; pinitol; sucrose

## 1. Introduction

Salinity is an important abiotic stress and a major threat to crop productivity worldwide. The soil area affected by salt stress is estimated to be over 1030 million hectares in the world [1]. In Canada, approximately 30% (20 million ha) of agricultural land either openly shows salinization or is at risk of becoming salinized [2,3]. In addition, soil salinization problems will likely worsen in the context of climate change, owing to the predicted increases of temperature and changes in precipitation patterns

worldwide [4]. The increasing use of poor-quality water for irrigation is an additional factor that exacerbates the global problem of soil salinity [5].

The adoption of crop rotations with deep-rooted perennial legumes is proposed as a way to decrease soil salinization [6]. The trend of soils to acidify following legume crops, or the effect of the root system of forage legumes on soil drainage and soil-nitrogen content, are among specificities making salt-tolerant legumes a potentially good option for soil-degraded soils [5]. Alfalfa (*Medicago sativa* L.) is one of the most important forage legume crops cultivated in the world owing to its high protein content and dinitrogen ($N_2$) fixation ability [7]. In Canada, alfalfa is the third-largest crop cultivated, accounting for 76% of the lands in Canadian Prairies [3,8].

Alfalfa is a glycophyte crop and a slight salinity stress of 20 mM NaCl could adversely affect its biomass yield [9]. Salt stress induces water and ion imbalances, and increases oxidative stress by the production of reactive oxygen species (ROS), which negatively alter photosynthesis, and hormonal and antioxidant balances causing a decrease in plant growth [10]. Plants can acclimate to salinity and osmotic stress according to different strategies including morphological, physiological, and biochemical changes in order to maintain an adequate water status, as well as good osmotic and ionic balances [10–13]. In response to salt stress, salt-tolerant plants accumulate organic solutes such as sugars and amino acids, and secondary metabolites that play important roles to improve salinity tolerance through osmotic adjustments and membrane protection [10,14].

Owing to the importance of alfalfa and the increasing incidence of environmental stresses, genetic improvement of alfalfa is strongly needed. The large genetic diversity of this species opens the door for improvement through conventional breeding [14]. However, breeding progress based on the selection of individual plants for their vigor in field nurseries has been limited in the past owing to the large within-field variability in salinity. To overcome these limitations, a facility for salinity testing under controlled conditions located in the Canadian prairies (Agriculture and Agri-Food Canada, AAFC, Swift Current salinity lab, SK; [3] was used for the recurrent selection of alfalfa within the two salt-tolerant cultivars Bridgeview and Halo.

Recurrent selection is a cyclical breeding method involving repeated exposures of a large number of genotypes to a given stress followed by the selection of genotypes with the most vigorous regrowth. The term genotype is used to identify the allelic composition of plants who have been selected for genes involved in salinity tolerance. After a few cycles of recurrent selection, the stress tolerance of various plant species has been significantly improved [15–19]. More specifically, alfalfa populations cyclically exposed to a stress-driven selection pressure have been shown to have an increased number of favorable alleles, leading to an improved stress tolerance [20].

In addition to plant adaptation, the surrounding microbiome could contribute to an improved stress tolerance. For instance, beneficial soil microorganisms such as rhizobia have been shown to alleviate salt stress and improve plant productivity in saline soils. Their beneficial effects have been reported to be linked to higher accumulation of osmolytes, such as proline and sugars [21,22]. The association of stress-tolerant cultivars and stress-tolerant rhizobia has been shown to result in a positive synergistic advantage in the ability of legumes to grow and survive under salt stress conditions [23,24]. We recently confirmed that the use of a salt-tolerant cultivar in symbiosis with a salt-tolerant rhizobium strain increased alfalfa yield under salt stress [9]. More specifically, a previous analysis of the biochemical changes in nodules in response to salinity stress showed that the accumulation of pinitol and sucrose [9], as well as of specific amino acids including proline, glutamate, ornithine, and aspartate [25], was linked to the salinity tolerance of alfalfa–strains associations, warranting further investigation of these specific osmolytes.

In the current study, we used the two initial cultivars tolerant to salt (TS0), Halo (H-TS0), and Bridgeview (B-TS0), along with two populations derived from these cultivars following three cycles of recurrent selection for salinity tolerance (H-TS3 and B-TS3). These four populations were inoculated with a salt-tolerant strain of *Sinorhizobium meliloti* (synonymous with *Ensifer meliloti*) and we compared their yield and physiological and biochemical responses to NaCl stress after three weeks'

exposure to five levels of NaCl concentrations. Our objectives were as follows: (1) to determine the extent of salt tolerance improvement after three cycles of recurrent selection in an alfalfa-*Sinorhizobium* symbiosis; (2) to identify physiological and biochemical changes associated with the improvement of salinity tolerance in alfalfa.

## 2. Material and Methods

(A) Recurrent selection for salinity tolerance in alfalfa

### 2.1. Plant Material

Two cultivars of alfalfa (*Medicago sativa* L.) improved for their tolerance to salt (Bridgeview and Halo) were used for the recurrent selection. Cultivar Bridgeview was developed at the Lethbridge Research and Development Centre, Agriculture, and Agri-Food Canada (Alberta, Canada), in collaboration with the Canada's Salinity Tolerance Testing Laboratory at Semiarid Prairie Agricultural Research Centre in Swift Current (Saskatchewan, Canada) [8,26]. Cultivar Halo was obtained from Cal/West Seeds [8].

### 2.2. Selection Protocol for Alfalfa Salinity Tolerance

The recurrent selection for salinity tolerance was conducted at the Canada's Salinity Tolerance Testing Laboratory according to the protocol of [2]. Recurrent selection was performed using the procedure illustrated in Figure 1. For both cultivars, each cycle of selection started with 520 genotypes (104 genotypes/tank × 5 tanks/cultivar) and ended up with the 50 best-performing genotypes to be intercrossed to generate a new population tolerant to salt (TS). Populations TS1 were obtained after one cycle, TS2 after two cycles, and TS3 after three cycles of recurrent selection.

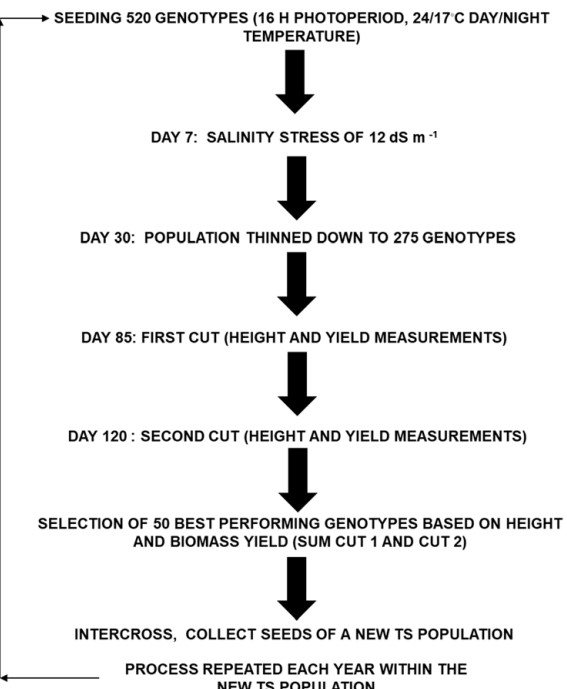

**Figure 1.** Schematic illustration of the procedure used for the recurrent selection of alfalfa for salt tolerance under controlled conditions. Successive cycles of recurrent selection were performed using two genetic backgrounds, cultivars Bridgeview and Halo. New populations tolerant to salt (TS) were obtained after each cycle.

In each tank, 104 genotypes were started from seeds in 0.709 m$^3$ tanks (0.95 m in diameter and 1.00 m high) filled with 0.645 m$^3$ of silica sand, allowing a growing surface area of 0.57 m$^2$ [8]. The ten growth tanks (five tanks per cultivar) were placed in an environment-controlled greenhouse according to a randomized complete block design. The day and night temperatures in the greenhouse were set at 24 °C and 17 °C, respectively, under an average photoperiod of 16 h, simulating a growing season between May and July in Swift Current (SK, Canada). Supplemental lighting from 475 W sodium lamps positioned 1.5 m above the silica sand surface with a photosynthetic photon flux density of 640 µmol photons m$^{-2}$ s$^{-1}$ was provided to extend the day length. A modified Hoagland's solution with an electrical conductivity (EC) of 1.5 dS m$^{-1}$ (2 mM Ca(NO$_3$)$_2$, 2.5 mM KNO$_3$, 1 mM MgSO$_4$, 0.50 mM NH$_4$NO$_3$, 0.17 mM KH$_2$PO$_4$, 0.05 mM KCl, 0.05 mM chelated Fe, and 0.023 mM H$_3$BO$_3$, plus micronutrients including Mn, Zn, Cu, and Mo) [27] was stored in a 612 L nutrient-brine supply reservoir and used for automatic surface fertigation. To induce a progressive salt stress, proportional solutions of 11.76 M CaCl$_2$ followed by 5.30 M MgSO$_4$ were added to the modified Hoagland's solution on two separate days to increase the EC to 5 dS m$^{-1}$; two days later, a solution of 31.92 M NaSO$_4$ was added to reach an EC of 10.5 dS m$^{-1}$; and four days later, a solution of 13.91 M NaCl was added to reach an EC of 12 dS m$^{-1}$ [8]. Growth tanks were flooded four times within a 24 h period with 4 h intervals between irrigation [8]. The final salt concentration was equivalent to 80 mM NaCl, representing a moderate salinity rating in the field. Sulfate-based salts were chosen to replicate saline conditions in Western Canada.

To ensure uniform sowing and to limit crowding as plants mature, the number of genotypes was thinned down to 55 plants per tank after emergence was completed (one month), representing a plant density of 96 genotypes m$^{-2}$. Two harvest cuts were performed during the growing period and the height and dry weight of individual genotypes were recorded. Around 120 days after sowing, the 50 best productive genotypes per cultivar (10 plants per growth tank) were selected based on their height and dry weight (sum of cuts 1 and 2) (Figure 1).

The 50 best-performing genotypes within each cultivar were transplanted into a 50:50 soil/soil-less peat mix, irrigated with water only and intercrossed to generate a new TS population. Seeds were harvested when plant maturity was reached and were used to undertake a new recurrent selection cycle. The two initial cultivars Halo (H-TS0) and Bridgeview (B-TS0) and the two populations obtained after three cycles of recurrent selection for salt tolerance within each of these two cultivars (H-TS3 and B-TS3) were used as material for the rest of the study.

(B) Assessment of salinity tolerance of contrasting alfalfa populations under controlled conditions

### 2.3. Inoculum Production

The salt-tolerant *Sinorhizobium meliloti* strain RM1521 isolated from Ottawa vicinity [28] was used to inoculate alfalfa seeds in this study. The growth conditions of RM1521 strain have been previously described by [9]. Briefly, RM1521 strain was grown in yeast mannitol broth placed in a shaking incubator (120 rpm, Lab-Line Orbit Environ-shaker, Melrose Park, IL) at 28 °C for one week. Then, the RM1521 inoculum was adjusted to 10$^9$ viable cells mL$^{-1}$.

### 2.4. Evaluation of Salt Tolerance of Plants Under Controlled Conditions

This experiment was conducted at the Québec Research and Development Centre of Agriculture and Agri-Food Canada. Pots (20 cm diameter, 20 cm deep) were filled with 900 g of Turface (Profile Products LLC, Buffalo Grove, IL), a reddish-tan calcined clay containing 60% SiO$_2$, 5% Fe$_2$O$_3$, and other chemicals of illite, moistened with 225 mL of water per L of Turface. A plastic saucer was placed under each pot to avoid contamination between treatments. Plastic pots and saucers were previously disinfected.

Seeds of the four alfalfa populations were surface-sterilized and washed with sterile water. Ten small cavities were made at the surface of each pot to facilitate seeding and inoculation. Two seeds

were planted per cavity and, at sowing, a 200 µL volume of RM1521 inoculum was applied in each cavity. Seeded pots were placed in plant growth chambers (Conviron, Winnipeg, Canada) with day/night temperatures of 22 °C/17 °C, and a 16 h photoperiod with a photosynthetic photon flux density of 400 µmol photons $m^{-2} s^{-1}$. Distilled water was applied once a day during the germination of seeds to avoid Turface dryness. Plants were thinned to 10 plants per pot one week after sowing. Then, a second 200 µL volume of RM1521 inoculum was applied.

After thinning, a daily fertigation (100 mL per pot) was applied using 0.25 × N-free nutrient solution for one week. At the third week of this study, 0.50 × N-free nutrient solution was applied until the end of the experiment (1 × N-free nutrient solution contains 111.80 mg P $L^{-1}$, 141.10 mg K $L^{-1}$, 2.10 mg Fe $L^{-1}$, 0.60 mg Mn $L^{-1}$, 0.12 mg Zn $L^{-1}$, 0.03 mg Cu $L^{-1}$, 0.39 mg B $L^{-1}$, 0.02 mg Mo $L^{-1}$, 48.62 mg Mg $L^{-1}$, and 0.95 mg Co $L^{-1}$). From the third week of the experiment, plants were also fertilized with N (2 mM N as $KNO_3$) and gradually subjected to salt stress induced by different NaCl concentrations (0, 40, 80, 120, or 160 mM NaCl). The salt stress was gradually increased during a week to avoid osmotic shock, as described in [9]. When each targeted NaCl concentration was reached, salt stress treatments were applied for two additional weeks. Plants were then harvested for yield measurement and physiological and biochemical assessments.

## 2.5. Physiological Assessment

Three plants per pot were gently harvested from the Turface and roots were washed with tap water. Excess water on roots was absorbed with paper towels. To limit within-pot variation, subsamples were collected on shoots and roots of the three plants and pooled. The fresh weights of shoot and root were recorded, and plant parts were dried at 55 °C to a constant weight for the total dry weight (DW) and water content (WC; g $g^{-1}$ DW) determinations.

The following equation was used to establish the relative WC in response to salt stress:

$$\text{Relative WC in response To Salt Stress (RWC} - \text{TSS; \%)} = \frac{\text{WC under NaCl stress}}{\text{WC at 0 mM NaCl}} \times 100 \quad (1)$$

Nodule location and abundance were assessed on these three plants by visual rating using a nodulation index (NI) modified from [29], at two root depths sections (distances from the crown from 0 to 7 cm, and from 7 to 20 cm) with the following scale: 0—no nodules, 1—few nodules, 2—numerous nodules, and 3—abundant nodules. This index was used as an indicator of the $N_2$-fixing potential of the alfalfa in symbiosis with the salt-tolerant strain of RM1521 in the four alfalfa populations in response to the five NaCl treatments.

## 2.6. Biochemical Analyses

Subsamples of leaves and nodules were collected on the remaining seven plants per pot, lyophilized, and ground for biochemical analyses. Leaves were ground using a mixer mill (Mixer Mill 301, Retsch Inc., Germany), while nodule samples were ground using a tissue homogenizer 3 × 30 s at 6800 rpm (Precellys 24, Bertin Technologies, France).

### 2.6.1. Carbohydrate Concentrations

For leaves, 0.20 g was extracted in 7 mL, while 0.04 g of nodules was extracted in 1.5 mL of a methanol/chloroform/deionized water solution with a ratio of 12:5:3 by volume. To stop enzymatic activity, tubes were heated at 65 °C for 20 min, then cooled in an ice bath, and kept at 4 °C overnight for optimal extraction. Tubes were homogenized, and leaves and nodules extracts were centrifuged (10 min at 2150× $g$ for leaves and roots and 3 min at 13,000 × $g$ for nodules). A 1 mL subsample of the supernatant was transferred into a 1.5 mL microtube and 250 µL of deionized water was added for phase separation. Microtubes were homogenized, decanted for 10 min, and centrifuged for 3 min at 13,000× $g$. Then, a 750 µL subsample of the supernatant was transferred to borosilicate tube and evaporated overnight at 43 °C using dry evaporation (Savant SpeedVac plus, Model SC210A, USA).

Samples were solubilized in 750 μL of deionized water, homogenized on a vortex mixer, and transferred into 1.5 mL microtubes prior to High Performance Liquid Chromatography (HPLC) analyses.

Mono-, di-, tri-, and tetra-saccharides were separated and quantified on a Waters analytic system controlled by the Empower II software (Waters, Milford, MA, USA). Sugars were separated on an HPX-87P column (Bio-Rad) at 80 °C with a flow rate of 0.5 mL min$^{-1}$ with water. Peak identity and quantity were determined for sucrose, glucose, fructose, and pinitol by comparison to standards (Sigma–Aldrich, Oakville, ON, Canada). Starch was quantified as described by [9].

### 2.6.2. Amino Acid Concentrations

The concentrations of free amino acids in leaves, roots, and nodules were analyzed by HPLC from the same extract used for water-soluble carbohydrate determination. Twenty-one amino acids were analyzed: alanine, arginine, asparagine, aspartate, glutamate, glutamine, glycine, γ-aminobutyric acid (GABA), α-aminobutyric acid (AABA), histidine, proline, methionine, lysine, serine, leucine, isoleucine, ornithine, phenylalanine, threonine, tyrosine, and valine. These amino acids were separated and quantified using Waters ACQUITY UPLC analytical system controlled by the Empower II software (WATERS, Milford, MA, USA) as described by [25]. The results from amino acid determinations were expressed as concentrations on a dry matter (DM) basis (μmol g$^{-1}$ DM). The total free amino acids was the sum of each individual concentration of the 21 free amino acids.

### 2.7. Statistical Analyses

Statistical analyses were conducted using the mixed procedure of SAS 9.3, version 2012 [30]. A total of 80 pots were used (5 NaCl concentrations (0, 40, 80, 120, or 160 mM NaCl) × 4 populations (H-ST0, B-ST0, H-TS3, and B-TS3) × 4 replications (1 replication = 1 pot containing 10 plants)). Data were analyzed according to a completely randomized experimental design and a three-way analysis of variance (ANOVA) was used to study the effect of NaCl, cultivars, recurrent selection, and their interactions on biomass yield, relative water content, nodulation index of roots, concentrations of carbohydrates in leaves and nodules, concentrations of total free amino acids in leaves and nodules, and concentrations of the 21 individual amino acids in nodules. The REPEATED statement was generally used to model the heterogeneity observed in the data. Residual analysis was performed to verify the assumptions underlying the model. When the normality assumption was not met, a log transformation was used. In this study, to get further insights on changes due to recurrent selection within each genetic background, between-factors effects were analyzed using univariate ANOVA. Pairwise comparisons were made using protected Fisher LSD (least significant difference) at $p = 0.05$.

## 3. Results

### 3.1. Plant Biomass

Shoot biomass yield, expressed on a DM basis, was reduced in both cultivars with an increase of NaCl concentration from 0 to 120 mM (Table 1; Figure 2A). Shoot DM yield differed significantly in response to recurrent selection (Table 1). At 40 mM, shoot DM yield was 30% higher in TS3 than in TS0, but the difference was not significant. At 80 mM NaCl, the shoot DM yield was 20% and 44% higher in B-TS3 and H-TS3 than in B-TS0 and H-TS0, respectively (Figure 2A). Shoot DM yield was also higher in TS3 than in TS0 populations under 160 mM NaCl. On average, the shoot DM yield was 33 % higher in both populations obtained after three cycles of recurrent selection for salinity tolerance than in initial cultivars. For roots, the DM biomass was also reduced with the increase in NaCl concentration from 40 to 120 mM NaCl, while recurrent selection had no effect (Table 1; Figure 2B).

**Table 1.** Analysis of variance with probabilities (*p*-values) of fixed effects (NaCl treatments, cultivars, recurrent selection) and their interactions on plant growth traits.

| Effects | Shoot Biomass | Root Biomass | RWC-TSS [1] | | Nodulation Index | |
|---|---|---|---|---|---|---|
| | DW [2] | DW [2] | Shoot | Root | Shallow (0 to <7 cm) | Deep Root (7 to 20 cm) |
| NaCl | *** | *** | * | ** | * | 0.167 |
| Cultivars | 0.789 | 0.099 | *** | 0.958 | 0.710 | 0.120 |
| NaCl × Cultivars | 0.739 | 0.932 | 0.073 | 0.964 | 0.744 | 0.405 |
| Recurrent selection | ** | 0.281 | 0.968 | ** | 0.710 | 0.593 |
| NaCl × Recurrent selection | 0.582 | 0.923 | 0.928 | 0.485 | 0.967 | 0.944 |
| Cultivars × Recurrent selection | 0.704 | 0.180 | 0.569 | 0.377 | 0.458 | 0.106 |
| NaCl × Cultivars × Recurrent selection | 0.088 | 0.921 | 0.979 | 0.843 | 0.694 | 0.324 |

*, **, ***: significant at $p \leq 0.05$, $p \leq 0.01$, and $p \leq 0.001$, respectively. [1] RWC-TSS: relative water content in response to salt stress; [2] DW: dry weight (g) of shoot and root biomass.

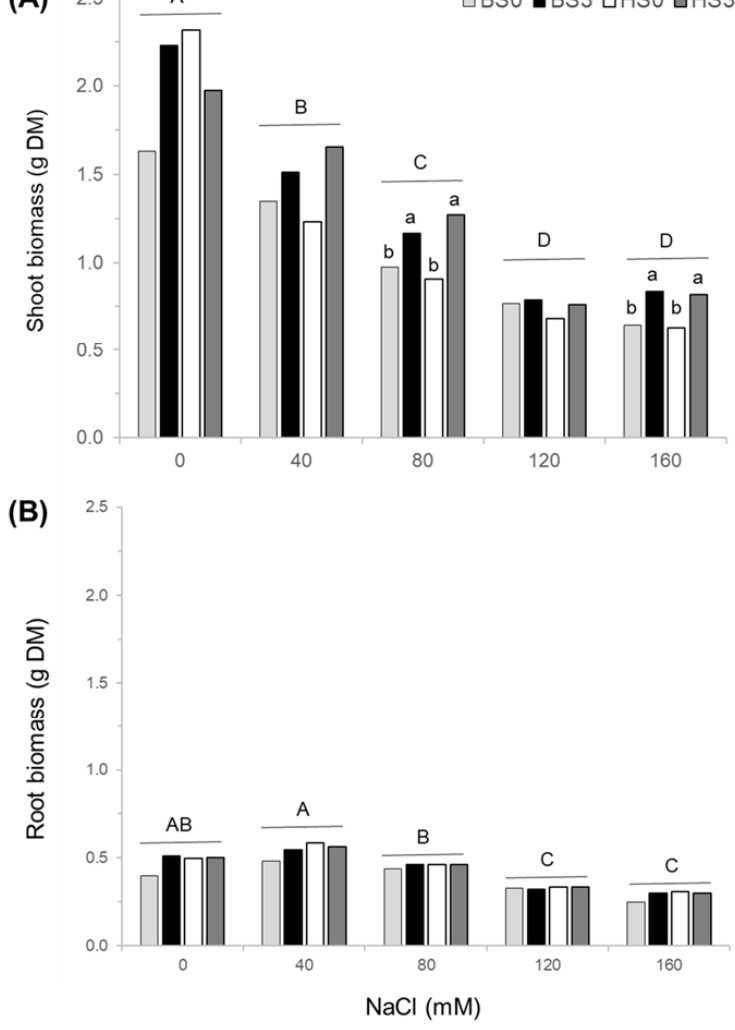

**Figure 2.** Shoot (**A**) and root (**B**) biomass yields of two cultivars of alfalfa (Bridgeview and Halo) exposed to different levels of NaCl (0, 40, 80, 120, or 160 mM NaCl). Dry matter (DM) yields of Bridgeview (B) and Halo (H) were evaluated in initial cultivars (TS0) and in populations obtained from these cultivars after three cycles of recurrent selection for salinity tolerance (TS3). Each bar represents the mean of four replicates. Different capital letters indicate significant differences between NaCl treatments, while lowercase letters indicate significant differences between populations in response to each NaCl treatment, as determined by the Fisher's least significant difference (LSD) test at $p \leq 0.05$. The absence of lowercase letters indicates that values are not significantly different.

### 3.2. Relative Water Content in Response to Salt Stress (RWC-TSS)

Relative shoot WC in response to salt stress differed significantly among NaCl concentrations, with a slight decrease with increasing NaCl concentration between 80 and 160 mM NaCl (Table 1; Figure 3A). There was a significant difference in relative shoot WC-TSS between cultivars with 35% to 56% higher values in Bridgeview than in Halo, while there was no effect of recurrent selection (Figure 3A). Conversely, relative root WC-TSS increased slightly with increasing salt concentration and a significant effect of recurrent selection was observed. At each salinity level, the relative root WC-TSS was significantly higher in TS3 populations of both cultivars than in initial TS0 populations (Table 1; Figure 3B). In Bridgeview, the relative root WC-TSS was 7%, 51%, 29%, and 39% higher in B-TS3 than in B-TS0 under 40, 80, 120, and 160 mM NaCl, respectively. In Halo, the relative root WC-TSS was 11%, 12%, 20%, and 11% higher in H-TS3 than in H-TS0 under 40, 80, 120, and 160 mM NaCl, respectively (Figure 3B).

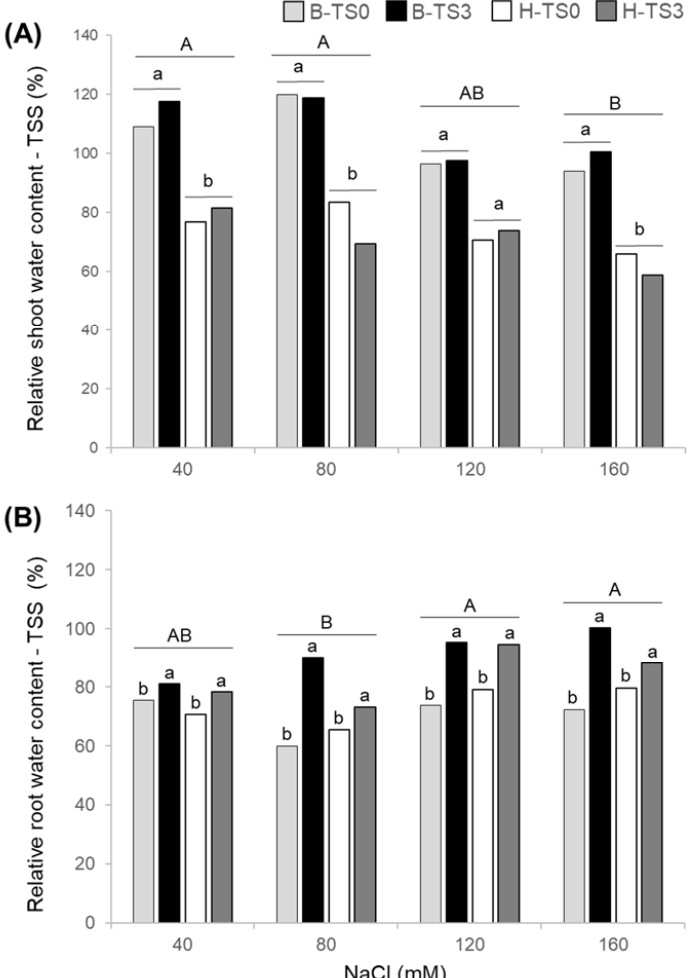

**Figure 3.** Relative water content (WC) in response to salt stress of shoots (**A**) and roots (**B**) of two cultivars of alfalfa (Bridgeview and Halo) exposed to different levels of NaCl (0, 40, 80, 120, or 160 mM NaCl). Relative WC in response to salt stress of Bridgeview (B) and Halo (H) was evaluated in initial cultivars (TS0) and in populations obtained from these cultivars after three cycles of recurrent selection for salinity tolerance (TS3). Each bar represents the mean of four replicates. Different capital letters indicate significant differences between NaCl treatments, while lowercase letters indicate significant differences between populations in response to each NaCl treatment, as determined by the Fisher's least significant difference (LSD) test at $p \leq 0.05$.

### 3.3. Nodulation Index of Roots

Nodulation was observed on the roots of all treatments and populations, even under a severe salt stress of 160 mM NaCl, as shown by the NI, which is based on nodule abundance. The NI of shallow roots (from 0 to 7 cm deep) decreased progressively with the increase in salt concentration from an average of 2.25 (~abundant) under 0 NaCl to 1.75 (~numerous) under 160 mM NaCl, while no effect of cultivars or recurrent selection was observed (Table 1; Figure 4A). The NI measured in deeper roots (from 7 to 20 cm deep) was not affected by the severity of salt stress and remained around 1.50 (between few and numerous) for all treatments (Figure 4B). At 160 mM NaCl, a significant effect of recurrent selection was observed in cultivar Bridgeview only, with a 75% higher NI in B-TS3 than in B-TS0 (Figure 4B).

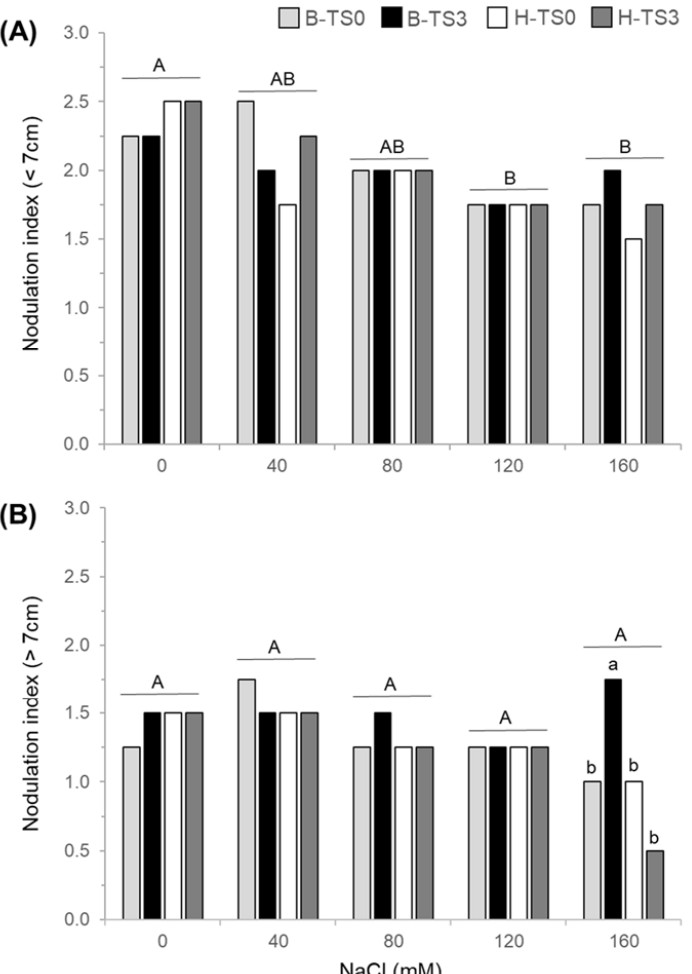

**Figure 4.** Nodulation index (NI) of potted plants measured in shallow (from 0 to 7 cm deep starting from the crown) (**A**) and deep roots (from 7 cm to 20 cm deep) (**B**) of two cultivars of alfalfa (Bridgeview and Halo) exposed to different levels of NaCl (0, 40, 80, 120, or 160 mM NaCl). The NI of Bridgeview (B) and Halo (H) was evaluated in initial cultivars (TS0) and in populations obtained from these cultivars after three cycles of recurrent selection for salinity tolerance (TS3). Each bar represents the mean of four replicates. Different capital letters indicate significant differences between NaCl treatments, while lowercase letters indicate significant differences between populations in response to each NaCl treatment, as determined by the Fisher's least significant difference (LSD) test at $p \leq 0.05$. Lowercase letters indicate that values are significantly different between populations at 160 mM NaCl.

### 3.4. Carbohydrate Concentration

Sucrose concentration in leaves of all populations was around 70% higher under moderate or severe NaCl stress (80, 120, and 160 mM NaCl) than in unstressed or mildly stressed plants (0 and 40 mM NaCl, Figure 5A). We observed a slight significant difference in sucrose concentration in leaves between cultivars with a higher concentration in Bridgeview (23.4 mg g$^{-1}$ DM) than in Halo (21.1 mg g$^{-1}$ DM). Except in unstressed plants (0 mM NaCl), sucrose concentration was around three times higher in nodules than in leaves (Figure 5A). In nodules, sucrose increased significantly with an increasing salinity level, but there was no difference between cultivars or recurrent selection (Table 2; Figure 5A).

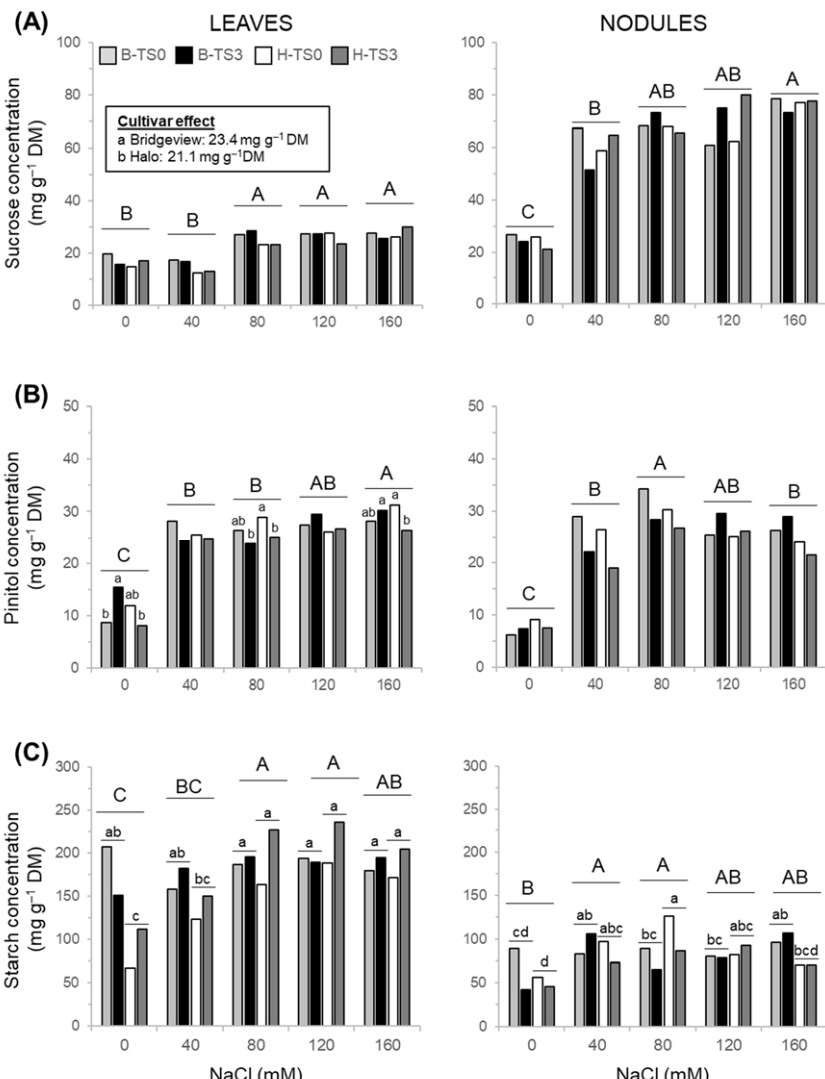

**Figure 5.** Concentrations of sucrose (**A**), pinitol (**B**), and starch (**C**) in leaves and nodules of two cultivars of alfalfa (Bridgeview and Halo) exposed to different levels of NaCl (0, 40, 80, 120, or 160 mM NaCl). Carbohydrate concentrations in Bridgeview (**B**) and Halo (**H**) were evaluated in initial cultivars (TS0) and in populations obtained from these cultivars after three cycles of recurrent selection for salinity tolerance (TS3). Each bar represents the mean of four replicates. Different capital letters indicate significant differences between NaCl treatments, while lowercase letters indicate significant differences between populations in response to each NaCl treatment, as determined by Fisher's least significant difference (LSD) test at $p \leq 0.05$. The absence of lowercase letters indicates that values are not significantly different.

**Table 2.** Analysis of variance with probabilities (*p*-values) of fixed effects (NaCl treatments, cultivars, recurrent selection) and their interactions on carbohydrate concentrations (mg g$^{-1}$ DM) in leaves and nodules.

| Effects | Sucrose | | Pinitol | | Starch | |
|---|---|---|---|---|---|---|
| | Leaves | Nodules | Leaves | Nodules | Leaves | Nodules |
| NaCl | *** | *** | *** | *** | *** | ** |
| Cultivars | * | 0.939 | 0.305 | 0.082 | 0.075 | 0.580 |
| NaCl × Cultivars | 0.425 | 0.947 | 0.170 | 0.152 | * | * |
| Recurrent selection | 0.823 | 0.713 | 0.238 | 0.118 | 0.067 | 0.106 |
| NaCl × Recurrent selection | 0.913 | 0.315 | 0.056 | 0.119 | 0.799 | 0.065 |
| Cultivars × Recurrent selection | 0.489 | 0.519 | * | 0.448 | * | 0.683 |
| NaCl × Cultivars × Recurrent selection | 0.484 | 0.679 | 0.079 | 0.936 | 0.666 | 0.233 |

*, **, ***: significant at $p \leq 0.05$, $p \leq 0.01$, and $p \leq 0.001$, respectively.

Pinitol concentration in leaves increased with increasing salt stress and a significant interaction between cultivars and recurrent selection was obtained (Table 2; Figure 5B). In unstressed plants (0 mM NaCl), pinitol concentration in leaves was significantly higher in TS3 compared with TS0 for cultivar Bridgeview only. Under 80 and 160 mM NaCl, pinitol concentration was lower in TS3 than in TS0 in cultivar Halo only (Figure 5B). In nodules, pinitol concentration increased with salt stress from 0 to 80 mM NaCl and then decreased under a severe salt stress of 160 mM NaCl. There was no significant difference between cultivars or recurrent selection (Figure 5B).

Significant interactions between cultivars and NaCl concentration as well as between cultivars and recurrent selection were observed for leaf starch concentration (Table 2). Under 0 mM NaCl, starch concentration was higher in the leaves of Bridgeview populations than in those of Halo, while it was similar between both cultivars treated with 40, 80, 120, or 160 mM NaCl (Figure 5C). Concentration of starch in leaves of the Bridgeview cultivar was not affected by salt concentration, whereas its concentration increased in Halo cultivar, especially between 0 and 80 mM NaCl. Starch concentration was lower in nodules than in leaves (Figure 5C) and a significant interaction between NaCl concentration and cultivars was also observed in nodules (Table 2). Starch concentration in the nodules of Bridgeview and Halo was variable according to the levels of salt applied with no effect of recurrent selection (Figure 5C). The highest concentration of starch was obtained in nodules of Halo with the addition of 80 mM NaCl compared with the populations without salt addition. For the Bridgeview cultivar, starch concentration in nodules was higher in response to 40 and 160 mM NaCl than in nodules without salt addition.

Glucose and fructose concentrations remained low for all treatments in both leaves (2.6 and 2.1 mg g$^{-1}$ DM for Bridgeview and Halo, respectively) and nodules (2.7 and 0.3 mg g$^{-1}$ DM for Bridgeview and Halo, respectively) (data not shown).

*3.5. Amino Acid Concentrations in Leaves and Nodules*

There was a significant interaction between NaCl concentration × cultivar for the total free amino acid concentration in leaves (Table 3). In both cultivars, the lowest concentration of total free amino acids in leaves was observed under 0 mM NaCl. For Bridgeview, total amino acid concentrations increased progressively with the intensity of salt stress (40 mM to 160 mM NaCl), while in Halo, the concentration was similar for all stress treatments. Across all salinity treatments, a significant effect of recurrent selection was shown by a 10% lower concentration of amino acids in TS3 than in TS0 populations in both cultivars (Figure 6A, Table 3). In leaves, proline was the amino acid showing the most marked increase in concentration in response to salt stress, increasing from 3.5 µmol g$^{-1}$ DM in non-stressed plants to 60.5 µmol g$^{-1}$ DM under a severe salinity stress of 160 mM NaCl (Supplemental Table S1).

**Table 3.** Analysis of variance with probabilities (*p*-values) of fixed effects (NaCl treatments, cultivars, recurrent selection) and their interactions on concentrations ($\mu$mol g$^{-1}$ DM) of total free amino acids in leaves, roots, and nodules.

| Effects | Total Free Amino Acids | | |
|---|---|---|---|
| | Leaves | Roots | Nodules |
| NaCl | *** | *** | *** |
| Cultivars | 0.206 | 0.175 | 0.293 |
| NaCl × Cultivars | * | 0.129 | 0.413 |
| Recurrent selection | * | 0.649 | 0.346 |
| NaCl × Recurrent selection | 0.878 | 0.291 | 0.663 |
| Cultivars × Recurrent selection | 0.236 | 0.696 | 0.211 |
| NaCl × Cultivars × Recurrent selection | 0.371 | 0.730 | 0.662 |

\*, \*\*\*: significant at $p \leq 0.05$ and $p \leq 0.001$, respectively.

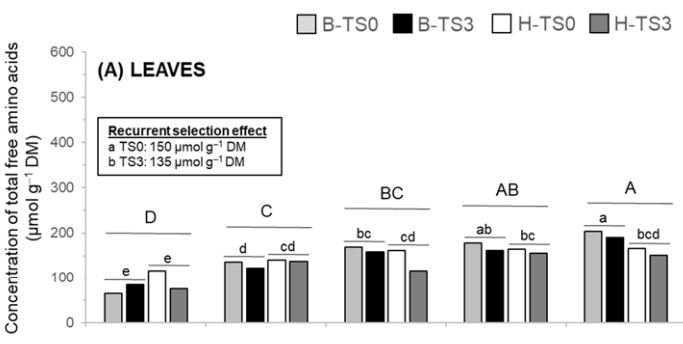

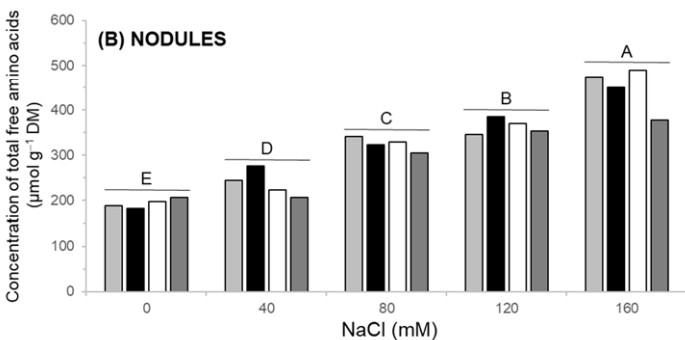

**Figure 6.** Concentration of total free amino acids in leaves (**A**) and nodules (**B**) of two salt-tolerant alfalfa cultivars (Bridgeview and Halo) exposed to different levels of NaCl (0, 40, 80, 120, or 160 mM NaCl). Amino acid concentrations in Bridgeview (B) and Halo (H) were measured in initial cultivars (TS0) and in populations obtained from these cultivars after three cycles of recurrent selection for salinity tolerance (TS3). Each bar represents the mean of four replicates. Different capital letters indicate significant differences between NaCl treatments, while lowercase letters indicate significant differences between populations in response to each NaCl treatment, as determined by Fisher's least significant difference (LSD) test at $p \leq 0.05$. The absence of lowercase letters indicates that values are not significantly different.

In nodules, the concentration of total free amino acids increased progressively with the concentration of NaCl from 200 $\mu$mol g$^{-1}$ DM with the 0 mM NaCl treatment to around 500 $\mu$mol g$^{-1}$ DM with the 160 mM NaCl treatment (Table 3, Figure 6B). Concentrations of individual amino acids increased with increasing NaCl concentrations in the nutrient solution from 0 to 160 mM NaCl, except for aspartate, alanine, glutamate, glutamine, and GABA, for which concentrations remained unchanged (Supplemental Table S1, Table 4).

**Table 4.** Concentration of amino acids ($\mu$mol g$^{-1}$ DM) in nodules of initial cultivars of alfalfa (TS0) and in populations obtained from these cultivars after three cycles of recurrent selection for salinity tolerance (TS3), exposed to 160 mM of NaCl. For each amino acid, precursors of the synthetic pathway are provided as well as putative metabolic functions related to salt stress tolerance. Significant NaCl and recurrent selection effects are indicated by arrows.

| Amino Acids | Precursors | Cultivars | | | | NaCl Effect [1] | Recurrent Selection Effect [1] | | Metabolic Function [2] | Other |
|---|---|---|---|---|---|---|---|---|---|---|
| | | Bridgeview | | Halo | | 0 to 160 mM | Bridgeview | Halo | | |
| | | TS0 | TS3 | TS0 | TS3 | | TS0 vs. TS3 | TS0 vs. TS3 | | |
| Glutamate (GLU) | GLU (from 2-oxoglutarate) | 41.93 [a] | 36.24 [a] | 41.17 [a] | 43.26 [a] | ↑ | = | = | Nitrogen assimilation and transport | |
| Proline (Pro) | GLU | 109.59 [a] | 110.22 [a] | 112.74 [a] | 99.43 [a] | ↑ | = | = | Compatible osmolyte, ROS scavenging | |
| GABA | GLU | 19.78 [a] | 16.18 [a] | 20.90 [a] | 21.25 [a] | = | = | = | ROS scavenging Coordinating carbon–nitrogen balance | Signaling |
| Arginine (Arg) | GLU | 16.05 [a,b] | 32.16 [a] | 19.22 [a] | 10.47 [b] | ↑ | = | = | Nitrogen storage and transport | |
| Glutamine (Glu) | GLU | 3.82 [a] | 3.92 [a] | 4.38 [a] | 3.55 [a] | = | = | = | Nitrogen assimilation and transport | |
| Ornithine (Orn) | GLU | 0.49 [a] | 0.52 [a] | 0.49 [a] | 0.37 [a] | ↑ | = | = | Polyamine synthesis | |
| Histidine (His) | GLU | 7.70 [a,b] | 9.34 [a] | 8.74 [a] | 6.07 [b] | ↑ | = | ↓ | Poorly investigated in plants | |
| Aspartate (ASP) | ASP (from oxaloacetate) | 4.11 [a] | 4.24 [a] | 5.58 [a] | 3.69 [a] | ↑ | = | = | Carbon skeletons | |
| Asparagine (Asn) | ASP | 209.24 [a] | 129.54 [a] | 187.14 [a] | 119.96 [a] | ↑ | = | = | Nitrogen assimilation, transport and storage | |
| Lysine (Lys) | ASP | 2.25 [b] | 5.14 [a] | 3.86 [a,b] | 2.05 [b] | ↑ | ↑ | = | | Signaling |
| Threonine (Thr) | ASP | 4.08 [b] | 7.35 [a] | 5.56 [a,b] | 4.01 [b] | ↑ | ↑ | = | Degradation to Val, Leu, Ile | |
| Methionine (Met) | ASP | 0.61 [b] | 1.95 [a] | 1.19 [a] | 0.90 [a,b] | ↑ | ↑ | = | Sulfate assimilation, polyamine synthesis Degradation to Val, Leu, Ile | |
| Isoleucine (Ile) | ASP/Pyruvate | 2.61 [b] | 8.66 [a] | 5.73 [a,b] | 3.01 [b] | ↑ | ↑ | = | Degraded into TCA cycle [3] | Branched-chain |
| Leucine (Leu) | ASP/Pyruvate | 2.72 [b] | 8.12 [a] | 4.94 [a] | 2.57 [a,b] | ↑ | ↑ | = | Degraded into TCA cycle | Branched-chain |
| Valine (Val) | ASP/Pyruvate | 4.63 [b] | 14.77 [a] | 10.37 [a,b] | 6.39 [b] | ↑ | ↑ | = | Degraded into TCA cycle | Branched-chain |
| AABA | GLU | 0.16 | 0.45 | 0.27 | 0.21 | ↑ | ↑ | = | Derivative of Ala Metabolite in Ile biosynthesis Catabolism Met, Tre, Ser | |
| Alanine (Ala) | Pyruvate | 19.29 [a] | 23.91 [a] | 26.64 [a] | 27.91 [a] | = | = | = | N assimilation and transport | |
| Tyrosine (Tyr) | Phosphoenol-pyruvate | 1.01 [b] | 3.79 [a] | 2.42 [a,b] | 1.34 [a,b] | ↑ | ↑ | = | Flavonoid and IAA synthesis [4] | Aromatic |
| Phenylalanine (Phe) | Phosphoenol-pyruvate | 2.06 [b] | 6.54 [a] | 4.02 [a,b] | 2.50 [b] | ↑ | ↑ | = | Flavonoid and IAA synthesis | Aromatic |
| Serine (Ser) | 3-Phospho-glycerate | 11.40 [a] | 13.81 [a] | 12.07 [a] | 9.79 [a] | ↑ | = | = | Carbon skeletons | |
| Glycine (Gly) | 3-Phospho-glycerate | 2.75 [a,b] | 3.74 [a] | 3.35 [a] | 2.47 [b] | ↑ | = | ↓ | Carbon skeletons | |
| Total amino acids (AAs) | | 473.97 [a] | 449.94 [a] | 489.52 [a] | 377.29 [a] | ↑ | = | = | | |

Each value represents the mean of four replicates. For each row, means followed by the same letter are not significantly different, as determined by Fisher's least significant difference (LSD) test at $p \leq 0.05$. GABA = $\gamma$-aminobutyric acid; AABA = $\alpha$-aminobutyric acid. [1] The arrows represent a significant increase or decrease of amino acid concentration according to NaCl effect or recurrent selection for each cultivar (TS0 vs. TS3), while an equal symbol indicates no significant effect. [2] Metabolic function for each group of amino acids. [3] TCA cycle: tricarboxylic acid cycle. [4] indole-3-acetic acid synthesis.

Under severe NaCl stress of 160 mM, a significant effect of recurrent selection was observed for 11 out of 21 amino acids (Table 4). For Bridgeview, concentrations of tyrosine, phenylalanine, isoleucine, leucine, valine, lysine, methionine, AABA, and threonine were higher in B-TS3 than in B-TS0. Conversely, three cycles of recurrent selection using the Halo cultivar significantly reduced the concentration of histidine and glycine as compared with H-TS0.

## 4. Discussion

### 4.1. Biomass and Relative Water Content

To assess the extent of salt tolerance improvement in response to recurrent selection in alfalfa, we compared the shoot and root biomass of two initial cultivars (TS0) and two populations obtained after three cycles of selection within these two cultivars (TS3), under five varying NaCl-stress intensities. Even though NaCl stress had, as expected, a negative impact on shoot yield, we obtained, overall, a 33% higher shoot yield in both TS3 populations compared with their respective TS0 populations when averaged across all levels of NaCl. The higher forage yield of TS3 populations under salinity stress clearly shows the effectiveness of our recurrent selection method performed indoors at improving salinity tolerance in alfalfa.

Plant growth combines complex physiological and biochemical processes, and a superior shoot growth could be related, as generally proposed, to better water availability, improved photosynthetic efficiency, increases in compatible osmolytes, and a lower transport of $Na^+$ ions leading to less desiccation and NaCl toxicity [4,10–13]. We observed that the relative shoot WC was not changed by recurrent selection, but that the relative root WC was higher in TS3 than in TS0 populations under NaCl stress. This indicates that one of the mechanisms of salt tolerance of recurrently selected populations is through a better adjustment of the root WC under NaCl stress. The higher relative root WC of TS3 populations translated into a higher shoot yield without an impact on shoot relative WC.

In a study on the physiological mechanisms of salt tolerance in alfalfa, [7] also reported that the improved salt tolerance of an alfalfa half-sib family compared with parental plants was the result of its ability to maintain a higher root WC under salt stress. In our experiment, root biomass decreased with increasing salt stress between 40 and 120 mM NaCl. However, root growth was less adversely affected by NaCl than shoot growth and there was no effect of recurrent selection on that trait. The increase in relative root WC with increasing salt stress could be an indication of a better capacity of a plant to maintain cell turgor pressure [10]. The root system acquires water and plant ions for plant growth and, as such, increases the water replacement rate to maintain cell turgor during salt-induced dehydration, and also excludes the $Na^+$ and $Cl^-$ from plant uptake [31].

The use of a salt-tolerant rhizobium strain could have further helped to maintain the root water balance [32]. Rhizobia can increase alfalfa salt tolerance through various mechanisms, including the prevention of excessive sodium uptake to maintain ionic balance in roots and leaves [33] and production of osmoprotective compounds [34]. It has, however, been reported that a high salinity level may negatively affect the nodulation capacity by the inhibition of initial steps of the establishment of *Rhizobium*-legume symbiosis [24,35]. The *Sinorhizobium meliloti* strain RM1521 has previously been shown to be salt-tolerant and to increase alfalfa capacity to withstand NaCl stress [9,25]. In the current study, we observed that increasing salt stress had little negative impact on root nodulation index, which could be partly explained by the salt tolerance of this rhizobium strain [9]. Inoculation of the pasture legume *Stylosanthes guianensis* with several strains of *Rhizobium* showed that root nodulation was not affected even under a 300 mM NaCl stress when this forage legume was inoculated with a salt-tolerant strain [34]. The rhizobium strain used in our experiment was able to infect and nodulate alfalfa under a severe salt stress of 160 mM NaCl, confirming the salt tolerance of that strain. However, the lack of uninoculated control or of the comparison with other strains prevent the drawing of a clear conclusion on the role of strain RM1521 in the nodulation performance of alfalfa in this study. At this high level of NaCl stress, we observed a higher nodulation index in B-TS3 than in B-TS0 for the 7–20 cm root

depth, showing that recurrent selection had a positive effect on the nodulation potential in cultivar Bridgeview. The presence of nodules in the lower part of the root indicated that *Sinorhizobium* had the opportunity to infect new root hairs. In addition to its positive effect on the plant water balance, a better nodulation under severe salt stress could have helped to maintain $N_2$ fixation, and thus partly explain the higher shoot biomass yield of TS3 populations under salinity stress.

## 4.2. Osmoprotectants in Leaves

To further characterize the underlying mechanisms of salinity tolerance in alfalfa, we proceeded with the biochemical analysis of compatible solutes in leaves and nodules, and compared improved-salt tolerant TS3 populations to initial TS0 cultivars. Compatible solutes are highly soluble organic compounds that could, under stress-induced cell desiccation, contribute to the cell osmotic adjustment without interfering with membrane integrity [13]. Sugars, polyols, and amino acids are non-toxic compounds that accumulate preferentially in the cytoplasm to maintain the osmotic balance and to protect the membrane structure to avoid membrane lysis and loss of semipermeability [36,37]. The sucrose concentration increased in leaves in response to increasing salinity stress, following a similar trend in all cultivars and populations regardless of their salinity tolerance. A marked increase in pinitol concentration, a cyclic sugar alcohol, was also observed in alfalfa leaves in the presence of salt. Pinitol has been reported to be involved in plant osmotic adjustment to maintain the metabolic activity under drought stress [38]. Transgenic *Arabidopsis* overexpressing a biosynthetic enzyme of the pinitol pathway demonstrated enhanced drought and salinity tolerance compared with the wild type [39]. It has been suggested that polyols could prevent oxidative damage, defend membrane integrity, and maintain enzyme activities [12,40]. Starch accumulated in leaves of Halo populations in response to salt stress, but remained at the same concentration in Bridgeview. In a review on the contribution of starch in plant fitness under stress, [41] reported that plants generally remobilize starch to provide energy and carbon under environmental conditions that limit photosynthesis. Here, starch in leaves did not decrease under salt stress, but we observed a marked increase of sucrose. Sucrose is a typical product of starch degradation that has been shown to stabilize membrane lipid bilayers during freeze-induced desiccation [42], and could mitigate the negative impacts of stress on alfalfa. We did not observe differences in carbohydrate accumulation between salt-tolerant populations TS3 and initial populations TS0. This result can be related to the fact that carbohydrate accumulation and catabolism is a basic mechanism of salt tolerance in alfalfa, as initial cultivars Bridgeview and Halo have also been selected for their salinity tolerance. On the other hand, this could indicate that our selection scheme performed indoor improved the salinity tolerance by exerting a selective pressure on other metabolic pathways and pyramiding these different mechanisms could have led to superior salt-tolerant TS3 populations [14].

Total free amino acids accumulated in leaves in response to salt stress, a result also observed in response to salt stress in different plant parts [36]. Among those amino acids, proline increased markedly in leaves from 4.0 µmol g$^{-1}$ DM in non-stressed plants to an average of 60 µmol g$^{-1}$ DM under a severe water stress of 160 mM NaCl, and was also the most abundant amino acid under salt stress (Supplementary Table S1). Proline has been shown to increase in response to salt tolerance in several plant species and to act as an osmoprotectant [7,36] as well as an ROS scavenger [12,43]. All these results support a previous study comparing different associations of alfalfa cultivars and rhizobium strains showing an increased concentration of compatible carbohydrates and amino acids in response to salinity concurring with a higher WC [9,25]. Thus, the recurrent selection scheme appears to have maintained the mechanisms of salt tolerance inherited from the initial cultivars.

## 4.3. Osmoprotectants in Nodules

As already mentioned, all alfalfa populations were inoculated with a salt-tolerant rhizobia strain, and we wanted to assess whether alfalfa-rhizobia symbiosis could have contributed to the increased salt tolerance to TS3 populations. Under salt stress, we observed a marked increase in nodule

sucrose concentrations, which reached concentrations twice as high as those in leaves. Salt stress also induced the accumulation of pinitol in nodules. Previous studies ([36,44]) also reported increases in concentration of sucrose and pinitol in alfalfa nodules in the presence of salt. Pinitol has been suggested to support nitrogenase activity in nodules [45]. The accumulation of soluble sugars has been related to higher survival of nodules under stress owing to an osmotic adjustment [44,46]. In addition to protecting nodules from salt-induced desiccation, the accumulation of compatible carbohydrates could have provided a sufficient carbon supply to nodules under stress to maintain a larger sink strength and N-fixing activity [9]. In a previous report, we observed a larger accumulation of pinitol in nodules of plants in association with the salt-tolerant strain RM1521 compared with a commercial strain. Here, the use of RM1521 with all alfalfa populations could have contributed to maintaining enough carbohydrates and was not linked with the higher salt tolerance of TS3 populations.

Amino Acids in Nodules under Severe Salt Stress

A general accumulation of free amino acids is usually observed in response to salt stress in different plant parts, including nodules [36]. Accordingly, we observed a marked increase of total free amino acids in nodules with increasing NaCl stress severity, reaching an average of 500 $\mu$mol g$^{-1}$ DM of nodules, a concentration two to three times higher than in unstressed nodules. Individual amino acids have been shown to increase in nodules under salt stress and to play specific protective roles. For instance, proline concentration doubled in nodules of alfalfa growing under 150 mM NaCl [47] and was reported to act as a precursor of polyamines that have radical scavenging properties [48]. Furthermore, interactions between plants and bacteria have been shown to naturally protect the plants from adverse conditions by synergistically modifying the concentration of stress-induced metabolites [33]. In that perspective, we decided to characterize further the accumulation of individual amino acids in plant nodules under a severe salt stress of 160 mM NaCl. The accumulation of most of the individual amino acids increased in the presence of salt stress, including amino acids of the glutamate pathway: glutamate, proline, arginine, histidine, and ornithine [47]. We also observed an increase in the aromatic amino acids (AAA) tyrosine and phenylalanine, which belong to the phosphoenolpyruvate pathway, and of methionine, threonine, isoleucine, leucine, and valine, which belong to the aspartate/pyruvate pathways, with the latter group including the branched-chain amino acids (BCAAs; isoleucine, leucine, and valine [49]). Accumulation of AAA and BCAA in Arabidopsis and rice leaves in response to drought was recently reported by [49], who proposed some clues to understand their role: BCAA could serve as compatible solutes or be used for respiration substrate, while AAA, as precursors of various secondary metabolites, would be involved in plant defense against pathogens. In legumes, the function of AAA and BCAA in source and sink organs, especially in nodules, is not well documented. However, it has been shown with pea (*Pisum sativum* L.) that the supply of BCAA to the bacteroids is regulated by the host plant to control nodule development and persistence [50]. The regulation of symbiosis and the protection of nodules are evolutionary mechanisms that are crucial for the host plant to ensure that the plant could resume the symbiosis and its growth as soon as the stress is alleviated. BCAAs have been proposed to fuel an alternative pathway of cellular respiration to provide energy to the plant under a water deficit [49,51]. As such, the accumulation of BCAA that we observed in response to NaCl stress could provide the additional energy needed to regulate ion transport and osmotic adjustment such as the exclusion-regulation of the NaCl transport from the soil to the root.

We observed that the effect of recurrent selection was more pronounced in Bridgeview than in Halo, as shown by a significant increase in concentrations of nine amino acids in nodules of B-TS3 compared with B-TS0 (Table 4). Notably, concentrations of lysine, AABA, threonine, methionine, BCAA, and AAA were higher in B-TS3 than in B-TS0. Even if speculative, this result suggests that the recurrent selection process in cultivar Bridgeview had an impact on the synthetic pathways of AAA and BCAA. Accumulation of BCAA under severe NaCl stress could also explain the higher nodulation observed in B-TS3 than in B-TS0, as the host plant has been shown to exert control on bacteroid

development through the BCAA transport to the nodule [50]. The recurrent selection of alfalfa for salinity tolerance appears to have selected different pathways to cope with salt stress according to each cultivar. Differences in adaptation strategies between cultivars have previously been reported for red clover [52].

## 5. Conclusions

The indoor method of recurrent selection was efficient to improve salinity tolerance in alfalfa, as shown by a 33% higher productivity obtained after three cycles of recurrent selection as compared with initial populations of two different genetic backgrounds. The higher relative root WC of populations obtained after three cycles of recurrent selection suggests that one of the mechanisms of salt tolerance of recurrently selected populations is through a better osmotic adjustment under salinity stress. Changes in osmolyte concentrations under salinity stress show that the two cultivars under study responded differently to recurrent selection for salinity tolerance. The differential adaptation of the cultivars to salinity could also be linked to their nodulation capacity under severe stress. For instance, under a 160 mM NaCl-stress, aromatic amino acids and branched-chain amino acids (BCAAs) increased in nodules of B-ST3 as compared with B-TS0, while these accumulations were not observed in H-TS3. BCAAs are known to control bacteroid development and their accumulation under severe stress could have contributed to the high nodulation of B-TS3.

Taken together, our results illustrate the multigenic nature of stress tolerance traits and show that our method of recurrent selection for salinity tolerance exerted pressure on different mechanisms depending of the genetic makeup of the two initial cultivars.

**Supplementary Materials:** The following are available online at http://www.mdpi.com/2073-4395/10/4/569/s1, Table S1: Concentrations of individual and total amino acids, and their means, in four alfalfa populations: Bridgeview (B-TS0) and Halo (H-TS0) and populations obtained after three cycles of selection within these two cultivars (B-TS3 and H-TS3) exposed to different levels of NaCl (0, 40, 80, 120, or 160 mM NaCl).

**Author Contributions:** All authors contributed to the conception of the work. C.G. performed the recurrent selection under salt stress. M.B. grew the four populations of alfalfa under salt stress and collected the data. A.B. and M.B. analyzed the data and drafted the manuscript. The paper was written by V.L. and revised by A.C., S.R., and G.F.T., A.B., F.P.C., and C.J.B. supervised the plants growth experiment; analyzed and interpreted the data; and modified, revised, and edited the manuscript. All authors have read and agreed to the published version of the manuscript.

**Funding:** The financial support of Centre SÈVE (strategic cluster of the "Fonds de recherche du Québec sur la nature et les technologies" (FRQNT) from the Quebec Government is gratefully acknowledged. The work was supported by a grant program of Agriculture and Agri-Food Canada (Project 1240).

**Acknowledgments:** The authors sincerely thank Josée Bourassa and Sandra Delaney for their technical assistance.

**Conflicts of Interest:** The authors declare no conflict of interest.

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
