# Peer review of "Physiological and Biochemical Responses to Salt Stress of Alfalfa Populations Selected for Salinity Tolerance and Grown in Symbiosis with Salt-Tolerant Rhizobium"

_agronomy, doi:10.3390/agronomy10040569_

Round 1

Reviewer 1 Report

The manuscript presents the results of a comparative analysis of the influence of NaCl-salinization on some physiological and biochemical parameters of two cultivars of alfalfa (as well as their generations of recurrent selection). The authors conducted an extensive and interesting work. The article was performed on a good methodological level with the using of modern methods. The results obtained undoubtedly deserve attention in view of their relevance. The article  subject to its minor revision in accordance with the comments of the reviewer.

General remarks:

  1. An explanation is needed why precisely these potential compatible osmolites have been studied? It is also known that Sinorhizobium meliloti can also synthesize glycine-betaine for protection against osmotic stress. An analysis of this compatible osmolyte would allow a better assessment of the contribution of the symbiont to plant protection.
  2. To understand how strain RM1521 helps plants, you need to have a test option without inoculation, or with inoculation with other strain. Otherwise, we can only assume positive effects in the conditions of this experiment. It is advisable to soften the phrase about the protective role of the strain.

Specific Notes:

Line 20 Higher relative root water content in TS3 populations indicated that root osmotic adjustment is one of the mechanisms of salt tolerance – one does not follow from the other.

Line 31. «Salinity is an important osmotic stress» – not quite true, this is stress combining both osmotic stress and toxic stress.

Line 156. Turface - what is this substrate? Based on what? What is its approximate composition?

Line 184. Relative water content is not a good term. Such a term is used for another important physiological parameter. Originally it estimates the current water content of the sampled leaf tissue relative the maximal water content it can hold at full turgidity. It is advisable to make a different term for your parameter, so as not to confuse the reader.

Additional materials must be fully translated into English.

Author Response

Reviewer 1:

   Comments and Suggestions for Authors

   The manuscript presents the results of a comparative analysis of the influence

   of NaCl-salinization on some physiological and biochemical parameters of two

   cultivars of alfalfa (as well as their generations of recurrent selection). The

   authors conducted an extensive and interesting work. The article was performed

   on a good methodological level with the using of modern methods. The results

   obtained undoubtedly deserve attention in view of their relevance. The article

   subject to its minor revision in accordance with the comments of the reviewer.

   General remarks:

     An explanation is needed why precisely these potential compatible osmolites

     have been studied?

The choice of these potential compatible osmolites is now explained in the manuscript. It is now specified that: "More specifically, a previous analysis of the biochemical changes in nodules in response to salinity stress showed that the accumulation of pinitol and sucrose (Bertrand et al. 2015), as well as of specific amino acids including proline, glutamate, ornithine and aspartate (Bertrand et al. 2016) was linked the salinity tolerance of alfalfa-strains associations, warranting further investigation of these specific osmolytes.

It is also known that Sinorhizobium meliloti can also

     synthesize glycine-betaine for protection against osmotic stress. An analysis

     of this compatible osmolyte would allow a better assessment of the

     contribution of the symbiont to plant protection.

We agree with the reviewer that the analysis of glycine-betaine would give a good assessment of the contribution of the symbiont to plant protection. However, glycine-betaine was not quantified in this study. It could be include in a future study.

     To understand how strain RM1521 helps plants, you need to have a test option

     without inoculation, or with inoculation with other strains. Otherwise, we can

     only assume positive effects in the conditions of this experiment. It is

     advisable to soften the phrase about the protective role of the strain.

It is now specified that: "However, the lack of uninoculated control or of the comparison with other strains prevent to draw a clear conclusion on the protective role of strain RM1521 in this study".

   Specific Notes:

   Line 20 Higher relative root water content in TS3 populations indicated that

   root osmotic adjustment is one of the mechanisms of salt tolerance – one does

   not follow from the other.

"Indicated" as been changed for "suggests" to dissociate the two factors.

   Line 31. «Salinity is an important osmotic stress» – not quite true, this is

   stress combining both osmotic stress and toxic stress.

"Osmotic" has been changed for "abiotic" stress to better take into account the complexity of salinity

   Line 156. Turface - what is this substrate? Based on what? What is its

   approximate composition?

It is now specified in the manuscript that : "Turface is a reddish-tan calcined clay containing 60% SiO2, 5% Fe2O3 and other chemicals of illite".

   Line 184. Relative water content is not a good term. Such a term is used for

   another important physiological parameter. Originally it estimates the current

   water content of the sampled leaf tissue relative the maximal water content it

 can hold at full turgidity. It is advisable to make a different term for your

   parameter, so as not to confuse the reader.

We agree with the reviewer that Relative water content usually refers to full turgidity while in this manuscript it refers to the water content under 0 mM NaCl. The term was changes to : Relative water content (WC) in response to salt stress (TSS). The term "WC-TSS" has been added and well described and the calculation is given in the manuscript with enough details to avoid confusion.

   Additional materials must be fully translated into English.

The additional materials has been fully translated in English.

Reviewer 2 Report

In this work, the authors tested the overall performance of Alfalfa plants after three cycls of recurrent selection under salinity stress (TS-3) by comparing several growth as well as physiological parameters with the original populations (TS-0). In the manuscript, this comparison is described, and the higher performance of the TS-3 populations is shown and discussed.

The work was well performed and it fits the aim and scope of the journal Agronomy. The experiments were conducted rigorously and statistics were appropriate.

  • In the paragraph 2.2. “Selection protocol for alfalfa salinity tolerance”, the procedure of “recurrent selection” is described. Here I have some doubts about the use of the term “genotypes”: in fact, at the beginning of the paragraph, the authors talk about “genotypes”, but later on in the same paragraph (at lines 134-139) they talk about “plants”; then, again “genotypes” (line 140). Therefore, I suppose that the term “genotype” is used as synonym of “plant individuals”; am I right? If yes, I would suggest to replace the term genotypes with “plant individuals” or simply “plants”, to avoid confusion.
  • 37-38: indicate a reference for this statement
  • 49-51: indicate a reference for this statement
  • 76: it should be either “Rhizobium spp.” or “rhizobia”
  • 87 and following text: am I right, that the official scientific name of Sinorhizobium meliloti is now Ensifer meliloti?
  • 223: correct the superscript
  • 2, 3, 4, 5: add standard error bars
  • 3B, 80 mM NaCl: the bar of H-TS3 appears very similar to that of H-TS0; however, the post-hoc letter is “a” and not “b”, as I guess it should. Please, check and correct, if appropriate
  • 286-287: Adjust text font
  • 5C and 6A: capital letters for post-hoc test are missed here
  • 406: here it is stated that the average shoot biomass increase in TS3 populations was 20%. However, in the results and also in the conclusions it is stated 33%. Please, check and correct, if appropriate
  • 440: Rhizobia not in italics
  • 464: Arabidopsis in italics
  • I don’t like the conclusions; they are a summary of the significant results. In the conclusion paragraph, one would expect to read general statements about the significance of the study, as well as the potential impact and application.

Author Response

   Reviewer 2:

   Comments and Suggestions for Authors

   In this work, the authors tested the overall performance of Alfalfa plants after

   three cycls of recurrent selection under salinity stress (TS-3) by comparing

   several growth as well as physiological parameters with the original populations

   (TS-0). In the manuscript, this comparison is described, and the higher

   performance of the TS-3 populations is shown and discussed.

   The work was well performed and it fits the aim and scope of the journal

   Agronomy. The experiments were conducted rigorously and statistics were

   appropriate.

     In the paragraph 2.2. “Selection protocol for alfalfa salinity tolerance”, the

     procedure of “recurrent selection” is described. Here I have some doubts about

     the use of the term “genotypes”: in fact, at the beginning of the paragraph,

     the authors talk about “genotypes”, but later on in the same paragraph (at

     lines 134-139) they talk about “plants”; then, again “genotypes” (line 140).

     Therefore, I suppose that the term “genotype” is used as synonym of “plant

     individuals”; am I right? If yes, I would suggest to replace the term

     genotypes with “plant individuals” or simply “plants”, to avoid confusion.

The term genotype is used here to identify the allelic composition of plants who have been selected for genes involved in salinity tolerance. We made a thorough revision of the recurrent selection protocols and changed the terms accordingly. It has also been added L.59-60 that : "The term genotype is used here to identify the allelic composition of plants who have been selected for genes involved in salinity tolerance".

     37-38: indicate a reference for this statement.

The following reference has been added (Abiala et al. 2018)

     49-51: indicate a reference for this statement.

The following reference has been added : (Acosta-Motos et al. 2017).

     76: it should be either “Rhizobium spp.” or “rhizobia”

Changed for rhizobia

     87 and following text: am I right, that the official scientific name of

     Sinorhizobium meliloti is now Ensifer meliloti?

Reviewer 2 is right. However, both terms continue to be used in published scientific literature, with Sinorhizobium being the more common (Willems et al. 2003. International Journal of Systematic and Evolutionary Microbiology (2003), 53, 1207–1217). For this reason, we choose to keep the scientific name Sinorhizobium meliloti but we added in the manuscript that Ensifer meliloti is synonymous.

     223: correct the superscript

The superscript has been corrected

     2, 3, 4, 5: add standard error bars

To indicate significant differences between treatment, both approaches (standard error bars and letters) are accepted. To avoid graphs with duplicated information, we choose to present significant differences between NaCl treatments with different capital letters and significant differences between populations in response to each NaCl treatment with different lowercase letters ( as determined by the Fisher’s least significant difference (LSD) test at P ≤ 0.05).

     3B, 80 mM NaCl: the bar of H-TS3 appears very similar to that of H-TS0;

   however, the post-hoc letter is “a” and not “b”, as I guess it should. Please,

     check and correct, if appropriate

As indicated, the letter a is correct

     286-287: Adjust text font

Changed as suggested

     5C and 6A: capital letters for post-hoc test are missed here

The capital letters are now indicated in Figure 5C and 6A

     406: here it is stated that the average shoot biomass increase in TS3

     populations was 20%. However, in the results and also in the conclusions it is

     stated 33%. Please, check and correct, if appropriate

20% has been changed for 33%

     440: Rhizobia not in italics

Changed as suggested

     464: Arabidopsis in italics

Changed as suggested

     I don’t like the conclusions; they are a summary of the significant results.

     In the conclusion paragraph, one would expect to read general statements about

     the significance of the study, as well as the potential impact and

   application.

   The conclusion has been improved according to the suggestions of the reviewer.